# The UV–Vis spectrum of the ClCO radical in the catalytic cycle of Cl-initiated CO oxidation
Wen Chao [1], Robert Skog [2], Benjamin N. Frandsen [2,3], Gregory H. Jones[1,8], Kayla T. Pham [4], Mitchio Okumura [1], Mads P. Sulbaek Andersen[5,6], Carl J. Percival [7] & Frank A. F. Winiberg [7] ✉

In Venus's mesosphere, the observation/model discrepancy of molecular oxygen, $O_2$, abundance has been a long-standing puzzle. Chlorine atoms have been proposed as a catalyst to oxidize carbon monoxide through the formation of chloroformyl radicals (ClCO), removing $O_2$ and ultimately generating $CO_2$. However, relevant kinetic studies of this catalytic cycle are scarce and highly uncertain. In this work, we report the spectrum of the ClCO radical between 210–520 nm using a multipass UV–Vis spectrometer coupled to a pulsed-laser photolysis flow reactor at 236–294 K temperature and 50–491 Torr pressure ranges. High-level ab initio calculations were performed to simulate the observed spectrum and to investigate the electronic structure. In addition, we observed the formation of molecular chlorine, $Cl_2$, and phosgene, $Cl_2CO$, suggesting that both the terminal chlorine and the central carbon in the ClCO radical are reactive towards chlorine atoms. Most importantly, the reported spectrum will enable future measurements of essential kinetic parameters related to ClCO radicals, which are important in regulating the $O_2$ abundance in Venus's mesosphere.

Chorine (Cl) chemistry plays an important role in a wide range of industrial applications, including water treatment[1], drug discovery[2], development of photovoltaic materials[3,4] organic synthesis[5], and air pollution[6]. These processes involve the formation of Cl-containing intermediates. A well-known example is the catalytic destruction of the ozone layer by chlorine atoms in Earth's stratosphere, a phenomenon described in the work that led to the award of the 1995 Nobel Prize[7].

Beyond Earth, chlorine chemistry has been proposed to play a pivotal role in regulating the concentration of molecular oxygen in Venus's mesosphere[8]. The upper atmosphere of Venus (60−90 km, 300–150 K)[9] is composed primarily of $CO_2$[10,11], which is photolyzed ($CO_2 + h\nu \rightarrow CO + O$) by the ultraviolet (UV) radiation from the Sun. On the night side, O atoms recombine, emitting light from excited state $O_2$ molecules ($O + O + M \rightarrow O_2{}^* + M$, $O_2{}^*$ ($A^1\Delta_g$) $\rightarrow O_2$ ($X^3\textstyle\sum_g^-$) $+ h\nu$)[12]. However, previous telescope observations have shown that oxygen is essentially absent ($[O_2]/[CO_2] < 3 \times 10^{-7}$)[13], a result that cannot be resolved by current photochemical models, which overestimate oxygen concentrations, $[O_2]$, by more than a factor of ten near the upper cloud layer[14,15].

Direct oxidation of CO by $O(^3P)$ or $O_2$ proceeds by high reaction barriers and is too slow to impact $[O_2]$. As shown in Fig. 1, the chlorine-initiated catalytic cycle of CO oxidation through the formation of chloroformyl radicals (ClCO)[16] has been proposed to consume $O_2$ and generate peroxychloroformyl radicals (ClC(O)OO). The catalytic cycle is completed by reaction of ClC(O)OO with either Cl atoms, O atoms, SO or $SO_2$ (X in Fig. 1) to generate $CO_2$ and release the Cl atom back. To accurately estimate the effects of Cl atoms on CO oxidation, it is essential to have well-constrained kinetic parameters related to ClCO, paramount of which is the ClCO equilibrium constant ($K_{ClCO}$). To the best of our knowledge, studies of ClCO are scarce with only one study examining $K_{ClCO}$[17], determined by monitoring the disappearance of Cl atoms at various CO concentrations, with an uncertainty of $\pm$ a factor of three (2 standard deviations). This uncertainty leads to several orders of magnitude difference in the modeled $O_2$ levels ($10^9$–$10^{13}$ $cm^{-3}$) throughout the Venus upper atmosphere[15].

Direct spectroscopic and kinetic measurements of the ClCO radical would provide a unique pathway to accurately measure $K_{ClCO}$. Existing spectroscopic studies of the ClCO radical include an electron spin resonance measurement[18], and mid-infrared (IR) measurements both in an Ar-

[1]Division of Chemistry and Chemical Engineering, California Institute of Technology, Pasadena, CA, USA. [2]Department of Chemistry, University of Helsinki, Helsinki, Finland. [3]Aerosol Physics Laboratory, Tampere University, Tampere, Finland. [4]Department of Chemistry, Columbia University, New York, NY, USA. [5]Department of Chemistry and Biochemistry, California State University Northridge, Northridge, CA, USA. [6]Copenhagen Center for Atmospheric Research, Department of Chemistry, University of Copenhagen, Copenhagen, Denmark. [7]Jet Propulsion Laboratory, California Institute of Technology, Pasadena, CA, USA. [8]Present address: Department of Chemistry, University of Florida, Gainesville, FL, USA. ✉e-mail: frank.winiberg@jpl.nasa.gov

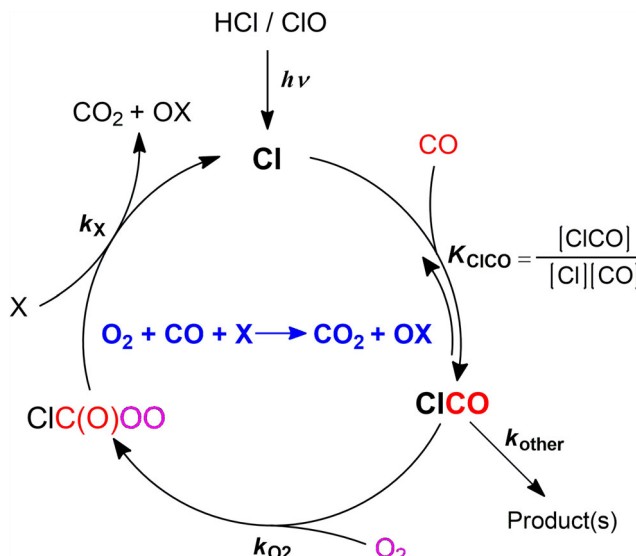

**Fig. 1 | The catalytic cycle of the Cl-initiated oxidation of CO in the Venus mesosphere.** X denotes the species proposed to be oxidized by the ClC(O)OO radicals in Venus's photochemical models (e.g., Cl, O, SO and $SO_2$).

matrix[19] and in the gas phase[20]. Despite theoretical studies that have predicted ClCO to have moderate absorption cross section in the UV–Vis range[21], direct measurements are still absent from the literature. Leveraging stronger cross sections in the UV–Vis range would allow direct detection of the ClCO radical with excellent signal-to-noise ratio to determine $K_{ClCO}$ with reduced uncertainty.

In this work, we have recorded the UV–Vis absorption spectrum of gas-phase ClCO using a multipass UV–Vis spectrometer coupled to a pulsed-laser photolysis flow reactor[22]. High-level ab initio calculations were conducted to elucidate the electronic structure and to simulate the recorded spectrum. Finally, chemical insights into the catalytic CO oxidation and the implications in the Venus mesosphere are discussed.

## Results and discussion
### The UV–Vis absorption spectrum of the ClCO radical
To investigate the ClCO spectrum between 270–510 nm, oxalyl chloride, $(ClCO)_2$, was photolyzed at 193 nm, producing two Cl atoms and two CO molecules with a yield of unity at 236 K and a total pressure of 50 Torr[23,24]. The available energy after the decomposition of $(ClCO)_2$ (~247 kJ mol$^{-1}$) is distributed as 13% in the Cl atoms and 87% in the CO molecules[25], leading to highly excited CO molecules. These hot CO molecules will be stabilized either through collision with the buffer gas or by IR emission[26]. Based on the reported forward and backward rate coefficients of the Cl + CO reaction ($k_+ = 3.2 \times 10^{-33}$ cm$^6$ s$^{-1}$ and $k_- = 1.5 \times 10^{-15}$ cm$^3$ s$^{-1}$)[17], we expected a prompt formation of the ClCO absorption signal peaking around 100 μs at 50 Torr of CO. A high $(ClCO)_2$ concentration (~$10^{16}$ cm$^{-3}$) was used to ensure a strong signal. However, a large fraction of the 193 nm photons were absorbed by $(ClCO)_2$, resulting in an estimated transmittance of 18%, based on the absolute cross section of $(ClCO)_2$ ($\sigma(ClCO)_2$(193 nm)[27] = $3.8 \times 10^{-18}$ cm$^2$) and the length of the flow reactor, excluding the purge region ($L = 45$ cm). This absorption resulted in inhomogeneous radical formation down the length of the flow reactor.

Figure 2A shows the recorded spectra of the $(ClCO)_2$ photolysis system in pure $N_2$ and CO at a total pressure of 50 Torr and at 236 K. The negative signal below 380 nm was attributed to the photodepletion of $(ClCO)_2$, $\Delta(ClCO)_2$. The introduction of CO produced a new spectral feature, characterized by two series of vibrational progressions in the 360–475 nm range. A significant deviation was observed from the $\Delta(ClCO)_2$ near 275 nm. The origin of this deviation is unclear. Although direct $Cl_2$ formation has been proposed from the photolysis of $(ClCO)_2$, the $Cl_2$ quantum yield for 193 nm

photolysis is likely smaller than 5%[24,28,29]. Moreover, $Cl_2$ has a maximum absolute cross section near 330 nm[30,31], which is far from 275 nm. Any influence from the direct photolysis products of $(ClCO)_2$ were eliminated by subtracting the recorded spectrum in the presence of CO from that in the presence of $N_2$. Additionally, the overall spectral profile is supported by the agreement between the experimental results and theoretical simulations (see Fig. 3).

We also investigated the ClCO spectrum in the 210–320 nm range by photolyzing molecular chlorine, $Cl_2$, at 351 nm. Figure 2B shows the recorded spectra of the $Cl_2$ photolysis system in pure $N_2$ and CO at a total pressure of 50 Torr and at 236 K. The negative absorption near 310 nm was attributed to the photodepletion of $Cl_2$, $\Delta Cl_2$, and was used to estimate the initial Cl atom concentration, $[Cl]_0$, for further analyses. The photolysis laser transmittance exceed 95% due to the small absorption cross section of $Cl_2$ near 351 m ($\sigma_{Cl_2}$(351 nm) = $1.9 \times 10^{-19}$ cm$^2$)[30,31], despite the high $[Cl_2] = 5.8 \times 10^{15}$ cm$^{-3}$. Therefore, we expected homogeneous radical formation along the flow tube[22].

As shown in Fig. 3, we assigned the absorption band with a vibrational progression near 380 nm and the structureless band peaking around 223 nm to the A-band and the B-band of the ClCO radical, respectively. In Fig. 2C, the B-band intensities, recorded at a delay time of 145 μs and 236 K, exhibit a rapid rise at low [CO] and reach a maximum at high [CO]. Since the observed signal intensities were proportional to the $[Cl]_0$, we normalized the signal intensities to $[Cl]_0 = (11.7 \pm 1.7) \times 10^{13}$ cm$^{-3}$, $(10.0 \pm 1.4) \times 10^{13}$ cm$^{-3}$ and $(8.9 \pm 1.3) \times 10^{13}$ cm$^{-3}$ at 498 Torr, 241 Torr and 91 Torr, respectively. At high [CO], the slow increase suggests that the ClCO → Cl + CO reaction is occurring as a result of the ClCO equilibrium.

By assuming rapid equilibrium, we can ignore the loss of Cl atoms. We derived formula (1) from the definition of the ClCO equilibrium constant and the mass balance relation, $[Cl]_0 = [Cl] + [ClCO]$, to analyze the observed CO dependence.

$$\frac{Abs_{ClCO}}{L_{eff}[Cl]_0} = \sigma_{ClCO} \frac{[CO]}{\frac{1}{K_{ClCO}} + [CO]} \quad (1)$$

The fitting in Fig. 2C yielded a $K_{ClCO}$(236 K) = $(1.5 \pm 0.2) \times 10^{-18}$ cm$^3$, which is 1.5 times smaller than the literature value[17], but still within the uncertainty ($1.2 \times 10^{-18}$ cm$^3 < K_{ClCO}$(236 K) $< 8.8 \times 10^{-18}$ cm$^3$). This analysis of the CO dependence provides a method to determine $K_{ClCO}$ from relative intensity measurements, thus avoiding errors in absolute concentrations or rate coefficients propagates into the uncertainty of $K_{ClCO}$.

Figure 2D shows the signal intensities of the A-band and B-band at 236–294 K. Fitting the data to the van't Hoff equation yielded an enthalpy change of $\Delta H = -(5.5 \pm 0.7)$ kcal mol$^{-1}$, which is 25% smaller than the literature. The experimental[17] and theoretical[32] enthalpy changes of the Cl + CO → ClCO reaction are $\Delta H°$(0 K) = $-(6.9 \pm 0.7)$ kcal mol$^{-1}$ and $-5.9$ kcal mol$^{-1}$, respectively. Both data sets show near identical temperature dependences, implying that the origins of both absorption bands are the same.

Figure 3 shows the ClCO spectrum obtained by subtracting the absorption spectra recorded with CO from those recorded with $N_2$. The A-band spectrum has a clear vibrational progression, and its overall shape closely matches the simulated spectrum. At wavelengths longer than 340 nm, the same vibronic progression was observed using 248 nm photolysis of both $(ClCO)_2$ and thionyl chloride $(Cl_2SO)$ as Cl atom precursors (Fig. S1). We note that interference from unknown species affected the spectra of the $Cl_2SO$ photolysis system (Fig. S3), and so we only use this data as a qualitative comparison for the spectral signatures. The vibronic progression contains a strong and a weak series, with average spacings of 409 and 408 cm$^{-1}$ (Fig. S2), respectively. A consistent difference of 215 cm$^{-1}$ between these two series between 340 and 440 nm suggests the presence of a vibrational mode with a frequency of 624 cm$^{-1}$.

The B-band absorption cross section was determined based on the observed CO dependence, while the A-band absorption cross section was derived by normalizing the absorption signal in the 270–300 nm window to

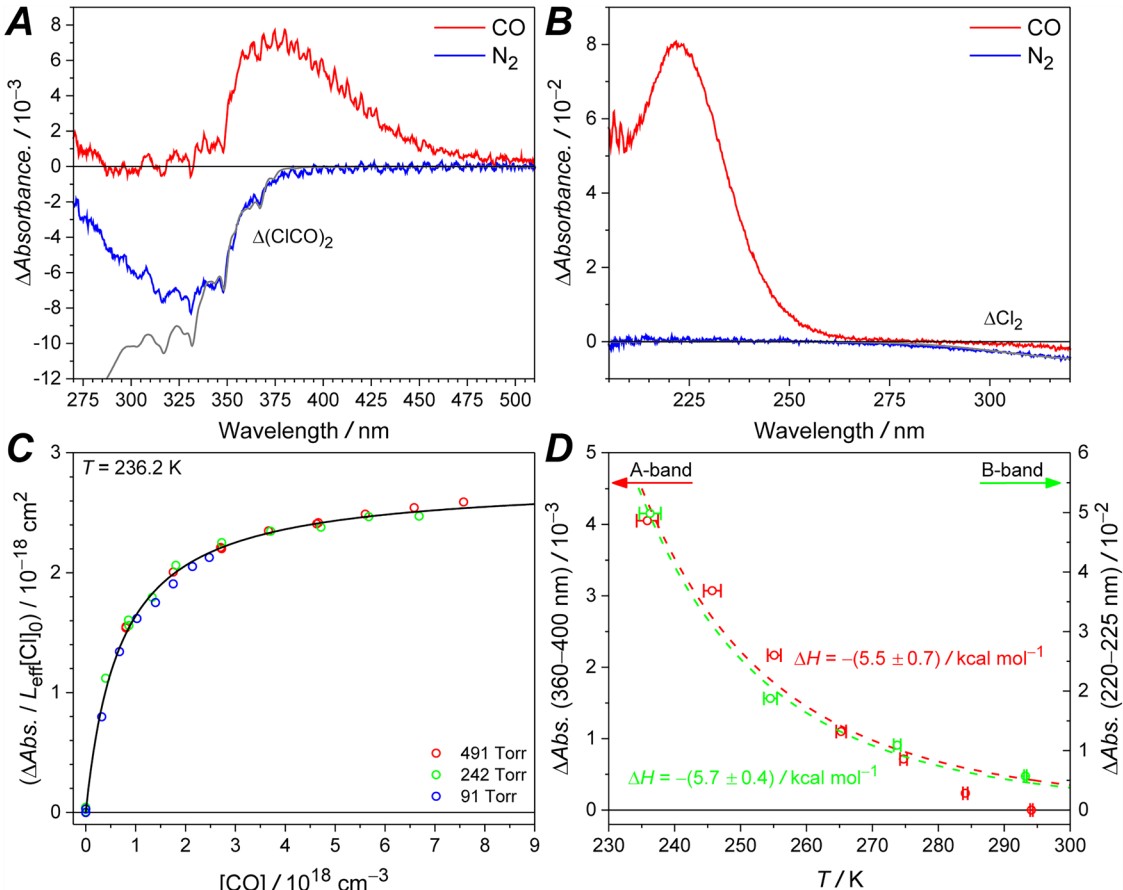

**Fig. 2 | The recorded spectra of the ClCO radical and the observed signal intensities as a function of CO concentration and temperature. A** Representative spectra of CO or N₂ as the buffer gas at a delay time 72 μs, 50 Torr and 236 K for an average of 12266 laser shots using 193 nm photolysis of (ClCO)₂. The gray line shows the contribution due to depletion of (ClCO)₂. **B** Representative spectra of pure CO or N₂ as the buffer gas at a delay time 145 μs, 50 Torr, and 236 K for an average of 6144 laser shots using 351 nm photolysis of Cl₂. The gray line shows the contribution due to depletion of Cl₂. **C** The normalized signal intensities in the 220–225 nm window

as a function of [CO] at a delay time 145 μs, 236 K, and different total pressures. The [Cl₂] at 491 Torr, 242 Torr, and 91 Torr are $7.5 \times 10^{15}$ cm⁻³, $6.9 \times 10^{15}$ cm⁻³, and $6.2 \times 10^{15}$ cm⁻³, respectively. The black line shows the fit to all data points. **D** The recorded signal intensities near 380 nm (red) and near 223 nm (green) as a function of temperature. The dashed lines show the fitting of the van't Hoff equation to all the data points in each dataset, yielding the estimated enthalpy change. The quoted error ranges are one standard deviation from the fit only.

the B-band profile. We determined the absolute cross section of $\sigma_{\text{ClCO}}(223\ \text{nm})$ to be ~$3 \times 10^{-18}$ cm². This value should be considered as a lower bound since complete scavenging of the Cl atoms is hard to quantify without kinetic analysis. Although the estimated absolute cross section is roughly two times smaller than theoretical predictions, the ratio of the predicted cross sections of the peaks of the B band and the A band, $\sigma(212\ \text{nm})/\sigma(345\ \text{nm}) = 6.6 \times 10^{-18}\ \text{cm}^2/2.5 \times 10^{-19}\ \text{cm}^2 = 26$, is consistent with the observed relative intensity of the peaks (~20) as shown in Fig. 3. For the uncertainty reported in this section, see Supplementary Information "Error Analysis".

## The electronic structure of the ClCO radical

Table 1 summarizes the ClCO calculation results. The geometry optimization yielded a ground state structure with a C—O bond length of 1.15 Å, a C—Cl bond length of 1.80 Å, and a bond angle of 129.1 degrees, consistent with previous calculations[20,21,33]. The harmonic vibrational frequencies are 367 cm⁻¹ for the bending mode, 606 cm⁻¹ for the CCl stretching mode, and 1963 cm⁻¹ for the CO stretching mode, which agree with previous IR measurements[19,20]. Four low-lying excited states were identified, with their orbital transition detailed in Table S1 and Table S2.

With an oscillator strength on the order of $10^{-2}$ and a vertical transition energy (5.91 eV, 210 nm) slightly larger than the observed B-band position, the $2^2A' \leftarrow X^2A'$ transition is the most likely candidate for the

stronger absorption band near 223 nm. On the other hand, the transitions to the $1^2A'$, $1^2A''$, and $2^2A''$ states have oscillator strengths on the order of $10^{-3}$ and vertical excited energies that lie between the peaks of the A-band and B-band, potentially contributing to the weaker absorption band near 360 nm.

The presence of vibrational progression on the red side of the A-band suggests the existence of a stable excited state. To gain further insights, we generated four representative slices of the potential energy surface (PES) of the ClCO radical along either the bending angle (∠ClCO) or the C–Cl bond length ($r$(C–Cl)), using MR-EOM-CC calculations with an "11 electron in 8 orbital CASSCF" reference averaged over 5 states.

Starting from the bent $X^2A'$ state geometry, Fig. 4B shows that the $1^2A''$ state and the $2^2A'$ state have stable energy minima near $r$(C–Cl) = 1.65 Å and 2.4 Å, respectively. Figure 4A shows that both states have energy minima at linear geometry and become degenerate to form the $1^2\Pi$ state, while the $X^2A'$ state becomes the $1^2\Sigma$ state. At linear geometry, Fig. 4D shows that a symmetry-allowed conical intersection[34] near $r$(C–Cl) = 1.7 Å between the bonded $1^2\Pi$ state and the repulsive $1^2\Sigma$ state. This conical intersection blurs the correlation between the $X^2A'$, $1^2A''$ and $2^2A'$ states and the $1^2\Pi$ and $1^2\Sigma$ states, as the $1^2A''$ state can become degenerate with either the $X^2A'$ state (Fig. 4C) or the $2^2A'$ state (Fig. 4A) depending on the $r$(C–Cl). As a result, the stabilities of the excited states are determined by their C–Cl bond length at the minimum energy geometry.

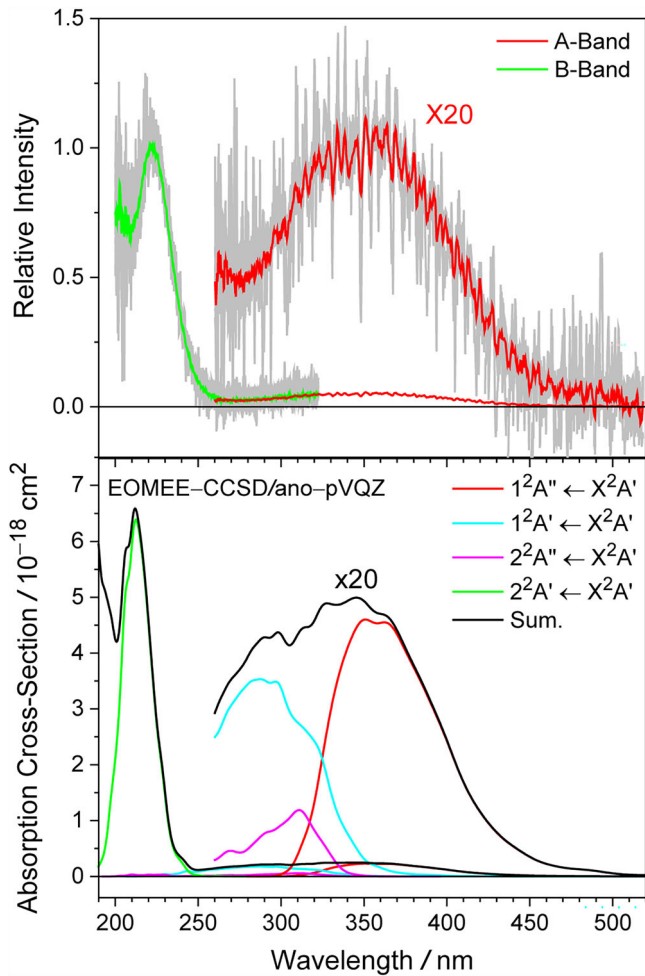

**Fig. 3 | The UV–Vis spectrum of the ClCO radical.** The upper panel shows the recorded A-band (red) and B-band (green) spectra of the ClCO radical at a total pressure of 50 Torr and at 236 K for an average of 12266 and 6144 laser shots, respectively. The lower panel shows the simulated spectrum (black) and the contributions from four low-lying excited states (red, cyan, magenta, and green). The simulated spectrum was constructed using the nuclear ensemble approach by sampling the ground state geometry based on a Wigner distribution with 5000 structures and calculating the vertical excitation energies and oscillator strength for each geometric structure at the EOMEE-CCSD/ano-pVQZ level of theory.

We conclude that the $2^2A'$ state is repulsive and the $1^2A''$ state primarily contributes to the observed vibrational progressions. Indeed, the simulated spectrum also shows that the red side of the A-band is dominated by the $1^2A'' \leftarrow X^2A'$ transition in Fig. 3. The appearance of the two series of vibrational progressions could be explained by the dependence of the excited state lifetime on the vibrational angular momentum, as observed in HCO and FCO (see Supplementary Information "Comparison with the HCO and FCO").

**The molecular orbital diagram and reactivity of ClCO**

The molecular orbital (MO) diagram (Fig. 5) summarizes the quantum calculations performed in this study. First, we considered the $3p$ orbitals of the Cl atom and the $1\pi$, $3\sigma$, and $2\pi$ orbitals of the CO molecule. At the linear geometry, the $3p$ orbital of the Cl atom along the molecular axis interacts with the $3\sigma$ orbital of the CO molecule. This interaction generates a pair of bonding ($\sigma$) and anti-bonding ($\sigma^*$) orbitals, characterized along the C–Cl bond. Additionally, the out-of-axis orbitals from the Cl atom and the CO molecule contribute to the formation of bonding ($\pi$), non-bonding ($\pi^n$), and anti-bonding ($\pi^*$) orbitals.

**Table 1 | The Geometries, Harmonic Frequencies, and Vertical Transition Energies of the ClCO Radical Calculated at the CCSD/ano-pVQZ level of theory**

| Geometry Optimized | | $X^2A'$ | | $1^2\Pi$ | |
|---|---|---|---|---|---|
| Geometry | $r$(C–O)/Å | 1.153 | | 1.192 | |
| | $r$(C–Cl)/Å | 1.796 | | 1.611 | |
| | ∠(ClCO)/degree | 129.07 | | 180 | |
| Frequency/cm$^{-1}$ | ClCO bending | 367.1 | | 411.4 | |
| | CCl stretching | 605.7 | | 733.7 | |
| | CO stretching | 1962.5 | | 2013.6 | |
| Vertical Transition Energy$^a$/eV | $1^2A'$ | 4.34 ($1.62 \times 10^{-3}$)$^b$ | | $1^2\Pi$ | 0.0 |
| | $2^2A'$ | 5.91 ($4.43 \times 10^{-2}$) | | $1^2\Sigma$ | 1.05 |
| | $1^2A''$ | 3.45 ($1.67 \times 10^{-3}$) | | | |
| | $2^2A''$ | 4.55 ($2.21 \times 10^{-4}$) | | $2^2\Pi$ | 6.91 |

$^a$Calculated at EOMEE-CCSD/ano-pVQZ level.
$^b$Oscillator strength is shown in parentheses.

We note that the energy gap between the $\sigma/\sigma^*$ pair increases as the C–Cl bond length decreases. Eventually, the energy of the $\sigma^*$ orbital can surpass that of the $\pi^*$ orbital as $r$(Cl–C) continues to decrease. When the $\sigma^*$ orbital energy is lower than the $\pi^*$ orbital energy, the unpaired electron prefers to occupy the $\sigma^*$ orbital, resulting in the repulsive $1^2\Sigma$ state. Conversely, occupation of the $\pi^*$ orbital results in the bound $1^2\Pi$ state as shown in Fig. 4D. The bending motion induces the mixing of the $\sigma^*$ and $\pi^*$ orbitals, lowering the energy of the singly occupied molecular orbital (SOMO) while increasing its anti-bonding character along the C–Cl bond. As a result, the stronger anti-bonding character of the SOMO relative to the $\pi^n$ orbitals makes the $1^2A'$ state and the $2^2A''$ state repulsive.

This MO diagram indicates that the SOMO correlates with the $2\pi$ orbital of the CO molecule, which has π anti-bonding character along the CO triple bond. Therefore, the CO bond strength in the ClCO radical is weaker than that of the CO molecule, as suggested by a smaller CO vibrational frequency[20]. The shape of the SOMO shows nearly equal character from the $3p$ orbital of the Cl atom and the $2\pi$ orbital of the CO molecule, as the EPR measurement[18] suggests a value of 0.42 for the unpaired electron density in the chlorine $3p$ orbital of the ClCO radical. This distribution of the unpaired electron implies that both the terminal chlorine and the central carbon will act as reaction centers in the ClCO radical.

As shown in Fig. 6, we have observed the reformation of $Cl_2$ at 49.9 Torr and at a delay time of 20 ms. Moreover, the reformation of $Cl_2$ is inhibited at 491.9 Torr while $Cl_2CO$ is observed. We are unable to fit the increase in absorbance at 260 nm with the ClCO and $Cl_2$ spectra alone, however inclusion of the $Cl_2CO$ spectrum allowed us to resolve this discrepancy. The formation of both $Cl_2$ and $Cl_2CO$ in our system indicates that both the terminal chlorine and central carbon are potential reactive centers in ClCO, as suggested by the MO diagram.

Photolyzing $Cl_2$ in the presence of CO at room temperature and one atmosphere pressure is a well-known recipe for synthesizing $Cl_2CO$, although studies on the reaction mechanism are sparse[35,36]. In our system, the Cl + ClCO reaction is the most plausible candidate for explaining the observed pressure effect on the formation of $Cl_2$ and $Cl_2CO$, as the Cl + ClCO → $Cl_2CO$ association reaction could be enhanced at higher pressures, while the chlorine extraction is pressure independent. In addition, the ClCO self-reaction is another potential candidate for the formation of $Cl_2$ and $Cl_2CO$. A weaker pressure effect may be expected if it is an elementary reaction not involving multiple steps or formation of energetically hot intermediates.

**The chemical insights and potential implications**

The catalytic role of Cl atom could be understood by the balance between energy gain and entropy cost in the formation of ClCO. The Cl–CO bond energy, $D_0$(Cl–CO) = 6.37 kcal mol$^{-1}$ (Table S3), is comparable to the

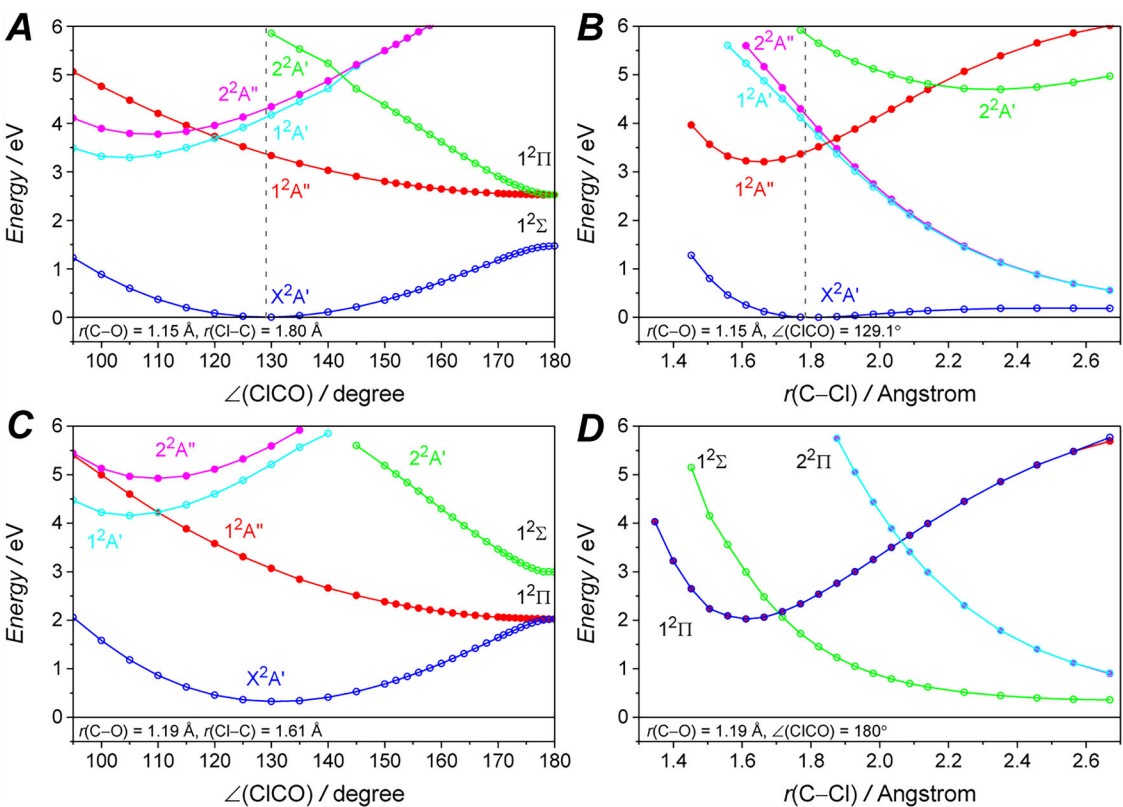

**Fig. 4 | The slices of the potential energy surface of the ClCO radical. A/C** and **B/D** shows the potential energy curve along the bending angle (∠ClCO) and C–Cl bond length ($r$(C–Cl)) with the rest degrees of freedom fixed at the X²A′ geometry and the 1²Π geometry for (**A/B**) and (**C/D**), respectively.

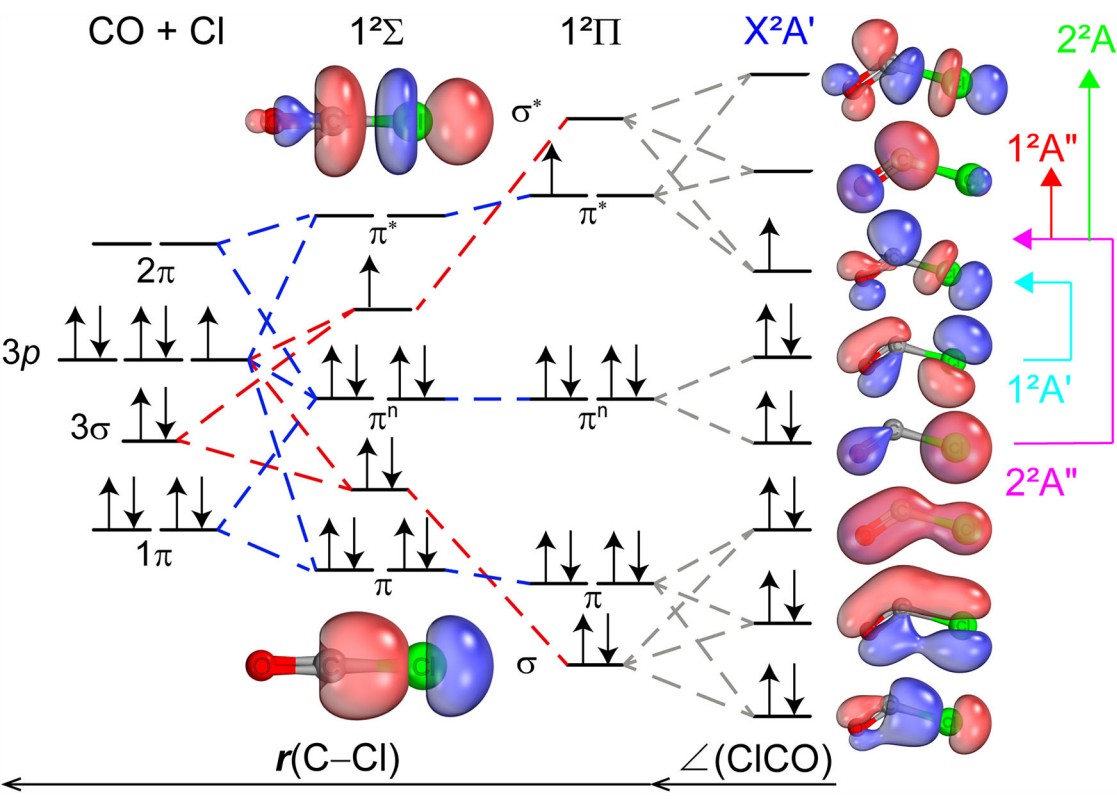

**Fig. 5 | Molecular orbital correlation diagram showing the valence electronic structure of the ClCO radical.** The $3p$ orbitals of the Cl atom and the $1\pi$, $3\sigma$, and $2\pi$ orbitals of the CO molecule were considered. The red dashed and blue dashed lines connect orbitals with $\sigma$ and $\pi$ characteristics along the C–Cl bond, respectively. The

gray dashed lines connect orbitals at linear and bend geometries. The vertical and horizontal arrows indicate the alpha and beta electron transitions from the X²A′ ground state, respectively.

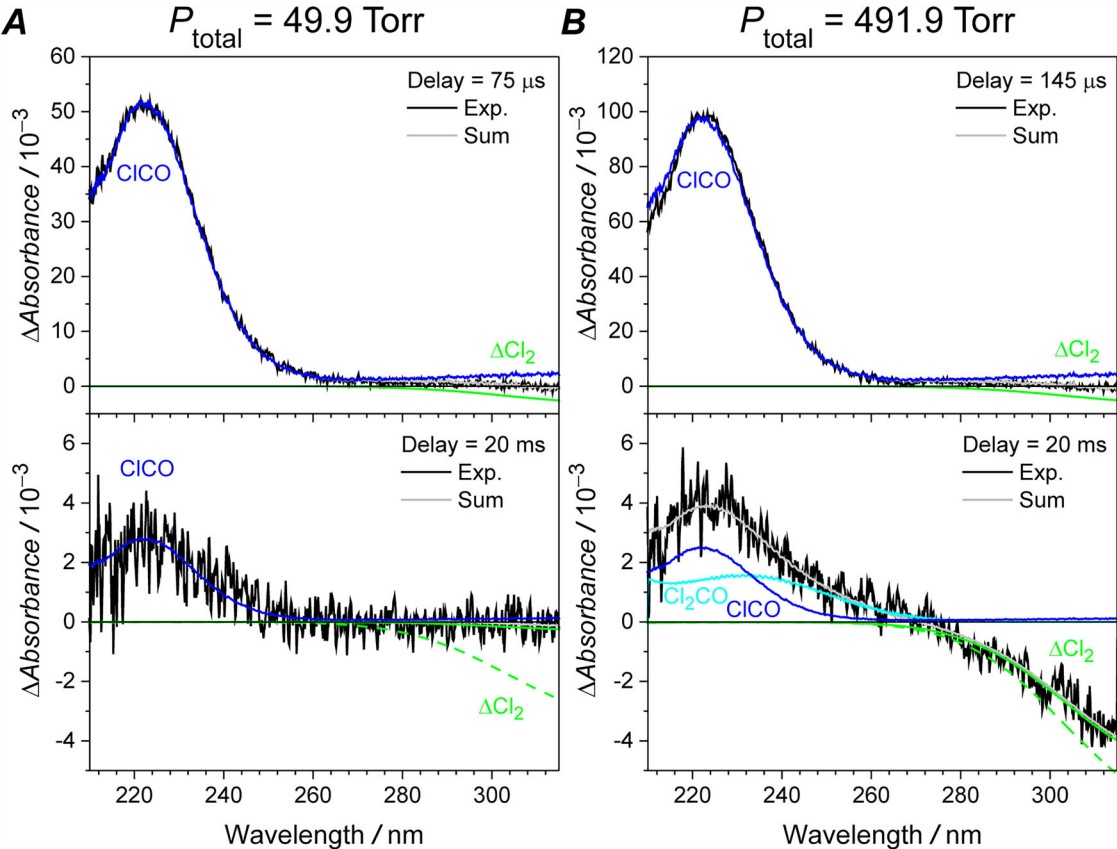

**Fig. 6 | The recorded spectra from the photolysis of Cl2 molecules at 351 nm in the presence of CO.** Experimental conditions at 236 K are $P_{total}$ = 49.9 Torr, $P(CO)$ = 49.9 Torr, $P(N_2)$ = 0 Torr, $P(Cl_2)$ = 83.7 mTorr for (**A**) and are $P_{total}$ = 491.9 Torr, $P(CO)$ = 45.6 Torr, $P(N_2)$ = 446.3 Torr, $P(Cl_2)$ = 181.6 mTorr for (**B**). The recorded spectra were deconvolved into the contributions of ClCO formation (blue, see the B band profile in upper panel of Fig. 3) and Cl_2 depletion[31] (green), except for the one recorded at a 20 ms delay time and 491.9 Torr, where phosgene[54] (cyan) was considered. The gray lines show the fitting results. The dashed green lines in both lower panels present the fitted $\Delta Cl_2$ from the corresponding upper panels.

entropy cost for bringing a water molecule to the reactants under typical atmospheric conditions[37]. In Earth's atmosphere, the energy gain from the formation of two hydrogen bonds ($\sim$7 kcal mol$^{-1}$)[38] is required to compensate the entropy cost for catalyzing the self-reaction of HO$_2$[39] and the reactions of Criegee intermediates with water vapor[40,41]. If the Cl-CO bond was too strong, a third reactant would be required to extract the Cl atom from the central carbon, preventing the release of a reactive Cl atom. For example, the F-initiated CO oxidation is not considered catalytic due to the stronger F–CO bond energy, $D_0(F–CO)$ = 22.3 kcal mol$^{-1}$[42,43]. On the other hand, if the Cl–CO bond was too weak, the concentration of ClCO would be too low to consume O$_2$ with a sufficiently fast rate. We posit that the Cl–CO bond energy is perhaps in a "Goldilocks zone", giving the highest catalytic efficiency analogous to the concept of the Sabatier principle.

Furthermore, this first direct observation of the ClCO radical in the UV–Vis region provides a new way to determine important parameters related to ClCO with reduced uncertainty at 236 K (e.g., $K_{ClCO}$ and the rate coefficient for the ClCO + O$_2$ reaction). The ClCO absorption signals near 223 nm over 236–294 K were strong, showing that the $K_{ClCO}$ could be determined from the observed CO dependence using Cl$_2$ as the chlorine precursor over 150–300 K for accurately modeling the catalytic effect of the Cl-initiated CO oxidation in Venus's mesosphere. A better estimation of the O$_2$ profile as a function of altitude is paramount, as it will be directly compared to measurements taken by the upcoming DAVINCI descent probe, scheduled for launch in the early 2030s[44,45].

## Conclusions
In this work, the UV–Vis spectrum of the ClCO radical was recorded at 236-294 K and 50-491 Torr using a multipass absorption spectrometer coupled

to a temperature-controlled flow reactor. A weak (A-band) and a strong (B-band) absorption band were observed, with peaks near 360 nm and 223 nm, respectively. The A-band shows two series of vibronic progressions, both with an average spacing of 409 cm$^{-1}$ and a shift of 215 cm$^{-1}$. Conversely, the B-band is structureless. The CO dependence of the B-band signal intensities yielded an absorption cross section of $\sigma_{ClCO}$(223 nm) near $3 \times 10^{-18}$ cm$^2$ and ClCO equilibrium constant of $K_{ClCO}$(236 K) = (1.45 $\pm$ 0.17) $\times 10^{-18}$ cm$^3$.

Four low-lying excited states were identified at the EOM-CCSD/ano-pVQZ level of theory. Representative slices of the ClCO PES were generated using the MR-EOM-CC method on a CASSCF (11,8)-5SA reference. The ClCO spectrum was also generated purely from theoretical calculations, and its overall profile agreed with the experimental measurements. These calculations showed that the 1$^2$A′, 2$^2$A′, and 2$^2$A″ excited states are repulsive. Moreover, the observed vibrational progression on the red side of the A-band is dominated by the 1$^2$A″ ← X$^2$A′ transition, while the blue side is dominated by the transitions to the 1$^2$A′ and 2$^2$A″ states. For the B-band, we assigned it to the 2$^2$A′ ← X$^2$A′ transition. A molecular orbital diagram was constructed to summarize the chemical insights obtained from the ab initio calculations. The shape of the SOMO suggests that both the terminal chlorine and the central carbon can act as reactive centers, which is supported by the observations of Cl$_2$ and Cl$_2$CO as the end products.

Most importantly, the reported ClCO spectrum suggests that direct monitoring of the formation and decay of ClCO radicals is feasible, which opens the door to measuring relevant kinetic parameters, such as $K_{ClCO}$ and the reaction rate coefficient of the ClCO + O$_2$ reaction, across a wide range of temperatures and pressures with reduced uncertainty. These data can help elucidate the catalytic role of Cl atoms in the Venus atmosphere and prepare atmospheric models for upcoming Venus's observation missions.

## Methods

### UV–Vis absorption measurements

For the experimental setup, a free-space broadband light source (LDLS, Energetiq EQ−99) was collimated using a parabolic mirror (Thorlabs, MPD149−F01, RFL = 101.6 mm, 90° OAP) and directed into a White cell for 10 passes, which gave an effective absorption length of $L_{\text{eff}} \approx 450$ cm. After exiting the multipass system, the light was focused into a spectrograph, installed with either a 600 grooves/mm or a 300 grooves/mm grating (Princeton Instruments, SpectraPro HRS−300). A half-high mirror within the spectrograph guided the bottom portion of the light to a photomultiplier tube (Hamamatsu R928) and the upper portion to an intensified CCD camera (Instruments PI-MAX4, 1024 × 256). A long-pass filter (Semrock LP02-257RU-30 × 40 or Rocky Mountain Instrument, $R_{\text{avg}} < 1\%$ at 200–280 nm, $R > 98\%$ at 351 nm) was placed at the exit of the White cell to isolate the probe beam from the photolysis excimer laser (Coherent Complex 205 F, ArF, KrF, or XeF). The experimental performance was evaluated previously[22].

The chlorine atom precursors, $(ClCO)_2$ (Sigma-Aldrich > 99%, ampule seal) and $Cl_2SO$ (Sigma- Aldrich > 99%), were delivered by a small stream of nitrogen flow in a bubbler, immersed into a temperature-controlled water bath (Fisherbrand, Isotemp 4100) at 292 K. The emerging flow was mixed with the CO (Airgas > 99.99%) and $N_2$ flows and was guided into a temperature-regulated double-jacket flow reactor. The gas mixtures were pre-cooled through a Graham condenser (Chemglass, CG-1830-30) with a cone shape exit, reducing the temperature inhomogeneity (236–298 K, with a margin of ±1.6 K) compared to our previous setup. To remove potential impurities, the CO stream was flowed through a potassium hydroxide (Baker Analzyed > 86.7%) trap immersed into a methanol ice slush. The $Cl_2$ (10% in He, Airgas) was used without purification. The pressures in the flow reactor (50–491 Torr) and precursor bubbler were continuously monitored using diaphragm gauges (MKS, 127AA series). The total pressure in the reactor was controlled by a throttle valve (MKS type 153).

### Theoretical methods

The CFOUR program suite[46] was utilized for performing various coupled cluster calculations, including single and double (CCSD) and higher order (CCSD(T) and CCSDT) computations. We selected the ano-pVXZ basis sets[47], as atomic natural orbital (ANO) basis sets have demonstrated exceptional effectiveness in calculating harmonic frequencies. The frozen-core approximation was applied since these basis sets were not designed for handling all-electron correlation. For ground state geometries, we optimized using CCSD with unrestricted Hartree-Fock for open-shell molecules and restricted Hartree-Fock for closed-shell molecules. The properties of excited states were calculated using the EOMEE-CCSD method.

While the EOM-CCSD method accurately treats dynamic correlation in the excited state, it failed in describing the behavior of adiabatic states near the conical intersection and during the bond-breaking process. Therefore, to qualitatively depict the PES and account for the complex interactions of the ClCO radical—particularly due to the Renner-Teller effect and pseudo-Jahn-Teller effect—we employed the multireference equation of motion coupled-cluster (MR-EOM-CC) method. This approach used a complete active space self-consistent field five states averaged (CASSCF(11, 8)−5SA) reference within the ORCA program suite[48]. For the visualization of orbitals, we used the IBOView package[49].

### Spectral simulations

For the spectral simulations we used the ground state geometry and normal modes calculated at the CCSD/ano-pVQZ level of theory using the CFOUR program suite. A nuclear ensemble, consisting of 5000 geometries, was created using a Wigner distribution based on normal mode displacements with the Newton-X (version 2.4 B06) program[50,51]. The large nuclear ensemble used minimizes the numerical integration error due to the statistical sampling in the final spectrum[52]. For each geometry vertical excitation energies and oscillator strengths for the ten lowest energy doublet excited states were calculated at the EOMEE-CCSD/ano-pVQZ level of theory with Gaussian 16 (Rev. C.02)[53] interfaced to Newton-X. The individual vertical transitions were convoluted with a 0.1 eV full width at half-maximum normalized Gaussian function, as recommended by Farahani et al.[52].

We tested different basis sets for the EOMEE-CCSD calculations and found that in all cases the final spectra were almost identical to the one calculated at the EOMEE-CCSD/ano-pVQZ level of theory. We also performed spectral simulations at three different temperatures, 0 K, 236 K, and 298 K. As can be seen in Figs. S4 and S5, the differences between the spectra at the different temperatures are negligible.

## Data availability

Supporting data about error analysis, analysis of the A-band vibronic structure, details of theoretical calculations are summarized in the supplementary information. Raw data of Figures shown in this work is available on the Caltech DATA repository at https://doi.org/10.22002/s58nj-d4j78.

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

## Acknowledgements

The experimental research herein was carried out at the Jet Propulsion Laboratory, California Institute of Technology, under contract with the National Aeronautics and Space Administration (NASA). Financial support was provided by the NASA Solar System Workings program. W.C. thanks for the fellowship support from the Josephine de Karman Fellowship Trust. R.S. thanks the Doctoral Programme in Chemistry and Molecular Sciences (CHEMS-DP) at the University of Helsinki for support. B.N.F. thanks for the fellowship support from the Carlsberg Foundation grant number CF22-0754. Computational resources for R.S. and B.N.F. were provided by the

Finnish IT Center for Science (CSC). B.N.F. and R.S. acknowledge Research Council of Finland Center of Excellence VILMA grant number 346369.

## Author contributions

W.C. performed all the experiments and analyzed the data. W.C. and G.H.J. worked on the MR-EOM-CC calculations. W.C. and K.T.P. recorded the ClCO B-band spectrum. R.S. and B.N.F. worked on the simulated spectrum. The corresponding author, F.A.F.W., makes the original idea and conceptualization. The draft was prepared by W.C. with further editing by W.C., R.S., B.N.F., M.P.S.A., C.J.P., F.A.F.W. All the laboratory resources were supported by F.A.F.W. and C.J.P. Computing resources for the HEAT and MR- EOM-CC calculations as well as the access of Caltech DATA repository were provided by M.O.

## Competing interests

The authors declare no competing interests.
