## [Transparent Peer Review file · Communications Chemistry]

The UV–Vis Spectrum of the ClCO Radical in the Catalytic Cycle of the Cl-Initiated CO Oxidation

Corresponding Author: Dr Frank Winiberg

Version 0:

Reviewer comments:

Reviewer #1

(Remarks to the Author)

The authors present direct UV-Vis absorption measurements of the radical ClCO, which plays a key role in facilitating CO oxidation on Venus. The absorption experiment utilizes a multipass spectrometer obtaining spectrum wavelength ranging from 210 to 520 nm. This ClCO absorption spectrum is obtained by two photolysis systems, each providing a distinct absorption range: one system uses (ClCO)₂ photolysis at 193 nm, covering the 270 to 520 nm range, while the other employs Cl₂ photolysis in CO at 351 nm, covering the 210 to 320 nm range.

The authors identified two major absorption bands for ClCO: the A-band, characterized by vibrational progression, suggesting it originates from a stable excited state, and the B-band, which is structureless and peaks at 223 nm, exhibiting significantly larger absorbance. High-level CCSD computations were performed to calculate the geometries and energy levels of the ground state and four low-lying excited states. The corresponding simulated UV-Vis spectrum shows excellent agreement with the experimental data.

Pressure- and temperature-dependent experiments were conducted to determine the absorption cross section for ClCO and the equilibrium constant (K_{ClCO}) for the reaction $\text{Cl} + \text{CO} \rightarrow \text{ClCO}$. The measured value of $K_{\text{ClCO}}(236 \text{ K}) = (1.5 \pm 0.2) \times 10^{-18}$ is in good agreement with literature values, but with a much lower uncertainty.

Overall, the manuscript is well-written and clearly motivated. The error analysis is thorough, and the experimental results are consistent with the computational findings, making the authors' claim of "direct measurement" highly convincing. I recommend publishing this manuscript in Communications Chemistry. However, I have a few questions and suggestions for minor revisions:

1. Line 71: In Figure 1, the variable "X" is not specified in the text, but it could be clarified in the figure caption. Additionally, I suggest hiding the arrows pointing from ClCO to Cl and CO, as they may distract from the emphasis on the CO oxidation catalytic cycle being presented here.
2. Line 76: Regarding the 193 nm photolysis of the (ClCO)₂ system, the depletion observed below 360 nm corresponds to the depletion of (ClCO)₂, which is used to estimate the initial Cl atom concentration, [Cl]₀. The Cl production follows the reaction $(\text{ClCO})_2 + h\nu \rightarrow 2\text{Cl} + 2\text{CO}$, which is actually a 2-steps dissociation process, with a Cl yield of 2 at 237 K. I was wondering if this is the only major dissociation pathway? A direct Cl₂ formation pathway has also been proposed for the (ClCO)₂ photolysis process (J. Phys. Chem. A 2017, 121, 2888–2895; Int. J. Chem. Kinet. 2024, 56, 482–498), with Cl₂ quantum yields ranging from 5% to 14%. Was this considered in your system? Since Cl₂ absorbs around 330 nm, it might affect the determination of the change in (ClCO)₂ and therefore the initial [Cl]₀.
3. Although [Cl]₀ are shown, I may have missed this, but I don't think the experimental concentrations of Cl₂ used in the Cl₂ photolysis experiments are specified like initial concentration of (ClCO)₂ (~10¹⁶ cm⁻³). This could be calculated by the readers with dissociation quantum yield, but might be convenient to have the number right away for better understanding of reactant composition.
4. Line 97 mentions that the photolysis laser transmittance was less than 5%, leading to homogeneous radical formation along the flow tube. I would have expected the opposite effect, where lower transmittance could lead to more localized photolysis.
5. Line 100: In Figures 2A and 2B, the gray lines are shown without description. It would be clearer to mention their

significance in the figure caption. Also, the fitting of the A-band (red dashed line) in Figure 2D is deviated for temperatures above 280 K. This is understandable given the low absorbance of the A-band at higher temperatures, but I wonder if the reported ΔH and its uncertainty (-5.5 ± 0.7) still account for these high-temperature data points?

6. Line 122: Consider changing “[CICO]” to “Abs_{CICO}” for greater clarity and to help readers connect this equation to Figure 2C.

7. Line 144: The meaning of the numbers in parentheses is unclear. Are they the intensity ratios of the B-band to the A-band?

8. Line 160: Consider including reference 19 when citing previous calculations related to the excited state dynamics.

9. Line 251: I just want to confirm my understanding: Are the blue curves in all four panels of Figure 6 the same as the curve shown in the upper panel of Figure 3 (stitching the green and red curves)?

Reviewer #2

(Remarks to the Author)

This is an important paper on the observation of the CICO radical in the gas phase by its UV-Vis absorption spectrum in conditions relevant to the chlorine chemistry in the Venus mesosphere. The manuscript is very well written and the results presented nicely and in a competent manner. High level ab initio electronic structure calculations are presented and help to understand the photochemistry of this important radical. The discussions and repercussions of the study are enlightening. In my view, the paper can be published as is.

Reviewer #3

(Remarks to the Author)

Review of The UV-Vis Spectrum of the CICO Radical in the Catalytic Cycle of the Cl-Initiated CO Oxidation by Chao W., et al.

Summary

This work reports on the UV-Vis spectrum of chloroformyl radicals (CICO), which are supposed to be an important atmospheric species in the carbon cycle in the upper Venus atmosphere, in the 236-294 K temperature and 50-491 Torr pressure ranges. The work presents ab initio calculations as well as laboratory measurements.

Recommendation

The paper is well-written and provides all the necessary information. I recommend publication with minor changes if the comments listed below are addressed.

Main comments

Line 84: What does T represent? Also, even though the provided value of L indicates it is the path length, I would specify the meaning of those symbols in the text.

Line 87: I see the negative values in Figure 2A below 380 nm, not 360 nm.

Figure 2A: What is the grey curve?

Line 116: What are the uncertainties on the [Cl]O signal intensities?

Line 126: Could you specify how much the temperature range matches with the literature on Venus's mesospheric temperatures? If I remember correctly, those vary between 150 K and 300 K. Could you add a discussion in the text on how those can be extended to cover the whole useful temperature range? This comment applies to all temperature dependence discussed in the paper...

Version 1:

Reviewer comments:

Reviewer #1

(Remarks to the Author)

The authors have carefully addressed all suggestions and comments. The issues raised in the previous version have been resolved, and the revised manuscript is now clear and accessible for readers. I highly recommend this manuscript for publication.

Reviewer #3

(Remarks to the Author)

I do not have further comments, all seems good to me. Thank you!

Reviewer #1 (Remarks to the Author):

The authors present direct UV-Vis absorption measurements of the radical ClCO, which plays a key role in facilitating CO oxidation on Venus. The absorption experiment utilizes a multipass spectrometer obtaining spectrum wavelength ranging from 210 to 520 nm. This ClCO absorption spectrum is obtained by two photolysis systems, each providing a distinct absorption range: one system uses (ClCO)₂ photolysis at 193 nm, covering the 270 to 520 nm range, while the other employs Cl₂ photolysis in CO at 351 nm, covering the 210 to 320 nm range.

The authors identified two major absorption bands for ClCO: the A-band, characterized by vibrational progression, suggesting it originates from a stable excited state, and the B-band, which is structureless and peaks at 223 nm, exhibiting significantly larger absorbance. High-level CCSD computations were performed to calculate the geometries and energy levels of the ground state and four low-lying excited states. The corresponding simulated UV-Vis spectrum shows excellent agreement with the experimental

data.

Pressure- and temperature-dependent experiments were conducted to determine the absorption cross section for ClCO and the equilibrium constant (K_{ClCO}) for the reaction $\text{Cl} + \text{CO} \rightarrow \text{ClCO}$. The measured value of K_{ClCO} (236 K) = $(1.5 \pm 0.2) \times 10^{-18}$ is in good agreement with literature values, but with a much lower uncertainty.

Overall, the manuscript is well-written and clearly motivated. The error analysis is thorough, and the experimental results are consistent with the computational findings, making the authors' claim of "direct measurement" highly convincing. I recommend publishing this manuscript in Communications Chemistry. However, I have a few questions and suggestions for minor revisions:

1. Line 71: In Figure 1, the variable "X" is not specified in the text, but it could be clarified in the figure caption. Additionally, I suggest hiding the arrows pointing from ClCO to Cl and CO, as they may distract from the emphasis on the CO oxidation catalytic cycle being presented here.

Reply: Thank for this suggestion. We agree that the double arrow is distracting and has been removed. However, we think the ClCO equilibrium should be highlighted, and a separated arrow has been added.

We have modified Figure 1 and specified "X" in the figure caption.

"... in the Venus mesosphere, where X denotes species (e.g. Cl atom, O atom, SO, and SO₂) proposed to be oxidized by the ClC(O)OO radicals in Venus's photochemical models.^{14,15}"

In addition, we have added a sentence to introduce the role of ClC(O)OO radical in the main text.

"As shown in Figure 1, the chlorine-initiated catalytic cycle of CO oxidation through the formation of chloroformyl radicals (ClCO)¹⁶ has been proposed to consume O₂ and generate peroxychloroformyl radicals (ClC(O)OO). The catalytic cycle is completed by reaction of ClC(O)OO with either Cl atom, O atom, SO or SO₂ (X in Figure 1) to generate CO₂ and release the Cl atom back."

2. Line 76: Regarding the 193 nm photolysis of the (ClCO)₂ system, the depletion observed below 360 nm corresponds to the depletion of (ClCO)₂, which is used to estimate the initial Cl atom concentration, [Cl]₀. The Cl production follows the reaction $(\text{ClCO})_2 + h\nu \rightarrow 2\text{Cl} + 2\text{CO}$, which is actually a 2-steps

dissociation process, with a Cl yield of 2 at 237 K. I was wondering if this is the only major dissociation pathway? A direct Cl₂ formation pathway has also been proposed for the (ClCO)₂ photolysis process (J. Phys. Chem. A 2017, 121, 2888–2895; Int. J. Chem. Kinet. 2024, 56, 482–498), with Cl₂ quantum yields ranging from 5% to 14%. Was this considered in your system? Since Cl₂ absorbs around 330 nm, it might affect the determination of the change in (ClCO)₂ and therefore the initial [Cl]₀.

Reply: Due to the small absorption signals obtained using (ClCO)₂ photolysis and the inhomogeneity of the radical formation, all [Cl]₀ reported in this work were estimated from the photolysis of Cl₂. We have removed the corresponding sentence to reduce confusion as below.

“The negative signal below 380 nm was attributed to the photodepletion of (ClCO)₂, $\Delta(\text{ClCO})_2$, ~~which was used to estimate the initial Cl atom concentration, [Cl]₀~~. The...”

We did not consider this direct Cl₂ formation pathway. As the reviewer mentioned, this channel might affect the determination of the change in (ClCO)₂, and therefore not only the initial [Cl]₀ but also the A band profile. However, we think that the impact from the direct Cl₂ formation is small for two reasons. First, this channel was reported for (ClCO)₂ photolysis at 248nm, 266nm, and 355 nm. At 193 nm, (ClCO)₂ is very likely to be excited to a distinct excited state, given the sharp increase in cross section with decreasing wavelength from 250 nm to 200 nm. Previous studies (J. Chem. Phys. 137, 164315, 2012) also reported a Cl atom yield of two at pressures of 25-100 Torr, indicating that the decomposition is fast. Second, any influence from direct products of (ClCO)₂ photolysis was cancelled in the difference spectrum.

We have modified this paragraph and added the following explanation to the text, including references, discussing the potential impact of the direct Cl₂ formation channel.

“The introduction of CO produced a new spectral feature, characterized by two series of vibrational progression within the 360 – 475 nm range. A significant deviation was observed from the $\Delta(\text{ClCO})_2$ near 275 nm. The origin of this deviation is unclear. Although direct Cl₂ formation has been proposed from the photolysis of (ClCO)₂, the Cl₂ quantum yield for 193 nm photolysis is likely smaller than 5%.^{24,28,29} Moreover, Cl₂ has a maximum absolute cross section near 330 nm,^{30,31} which is far away from 275 nm. Any influence from the direct products of (ClCO)₂ photolysis was eliminated by subtracting the recorded spectrum in the presence of CO from that in the presence of N₂. Additionally, the overall

spectral profile is supported by the agreement between the experimental results and theoretical simulations (see Figure 3).”

3. Although [Cl] _o are shown, I may have missed this, but I don't think the experimental concentrations of Cl₂ used in the Cl₂ photolysis experiments are specified like initial concentration of (ClCO)₂ (~10¹⁶ cm⁻³). This could be calculated by the readers with dissociation quantum yield, but might be convenient to have the number right away for better understanding of reactant composition.

Reply: Although we showed the [Cl₂] in Figure 6, we forgot to specify the [Cl₂] for Figure 2B and 2C. We have added [Cl₂] in the main text for Figure 2B, and in the figure captions for Figure 2C.

“The photolysis laser transmittance exceed 95 % due to the small absorption cross section of Cl₂ near 351 nm ($\sigma_{\text{Cl}_2}(351 \text{ nm}) = 1.9 \times 10^{-19} \text{ cm}^2$),^{30,31} despite the high [Cl₂] = 5.8 × 10¹⁵ cm⁻³.”

“(C) The normalized signal intensities in the 220 – 225 nm window as a function of [CO] at a delay time 145 μs, 236 K and different total pressures. The [Cl₂] at 491 Torr, 242 Torr and 91 Torr are 7.5 × 10¹⁵ cm⁻³, 6.9 × 10¹⁵ cm⁻³, and 6.2 × 10¹⁵ cm⁻³, respectively. The black line shows the fit to all data points.”

4. Line 97 mentions that the photolysis laser transmittance was less than 5%, leading to homogeneous radical formation along the flow tube. I would have expected the opposite effect, where lower transmittance could lead to more localized photolysis.

Reply: We really appreciate the reviewer's effort to find this mistake. The original sentence should be “the absorbance was less than 0.05”, which resulting in a transmittance greater than 95%. We have modified this sentence as below.

“The photolysis laser transmittance exceed 95 % due to the small absorption cross section of Cl₂ near 351 nm ($\sigma_{\text{Cl}_2}(351 \text{ nm}) = 1.9 \times 10^{-19} \text{ cm}^2$),^{30,31} despite the high [Cl₂] = 5.8 × 10¹⁵ cm⁻³. Therefore, we expected a homogeneous radical formation along the flow tube.²²”

5. Line 100: In Figures 2A and 2B, the gray lines are shown without description. It would be clearer to mention their significance in the figure caption. Also, the fitting of the A-band (red dashed line) in

Figure 2D is deviated for temperatures above 280 K. This is understandable given the low absorbance of the A-band at higher temperatures, but I wonder if the reported ΔH and its uncertainty (-5.5 ± 0.7) still account for these high-temperature data points?

Reply:

For the gray lines in Figure 2A and 2B, we have modified the figure captions.

“(A) Representative spectra of CO or N₂ as the buffer gas at a delay time 72 μ s, 50 Torr and 236 K for an average of 12266 laser shots using 193 nm photolysis of (ClCO)₂. The gray line shows the contribution due to depletion of (ClCO)₂.

(B) Representative spectra of pure CO or N₂ as the buffer gas at a delay time 145 μ s, 50 Torr and 236 K for an average of 6144 laser shots using 351 nm photolysis of Cl₂. The gray line shows the contribution due to depletion of Cl₂.”

For the fitting in Figure 2D, we did not exclude the data points above 280 K in the fitting. As a result, the reported uncertainty for the B band measurements (green) is larger than the A band measurements (red). However, the reported uncertainty is purely from the fitting results, which might be underestimated. We modified the figure caption as below.

(D) The recorded signal intensities near 380 nm (red) and near 223 nm (green) as a function of temperatures. The dashed lines show the fitting of the van't Hoff equation to all the data points in each dataset, yielding the estimated enthalpy change. The quoted error ranges are one standard deviation from the fit only.”

6. Line 122: Consider changing “[ClCO]” to “Abs_{ClCO}” for greater clarity and to help readers connect this equation to Figure 2C.

Reply: We thank the reviewer for pointing out this mistake. Equation (1) has been updated.

$$\frac{Abs_{ClCO}}{L_{eff} [Cl]_0} = \sigma_{ClCO} \frac{[CO]}{1/K_{ClCO} + [CO]}$$

7. Line 144: The meaning of the numbers in parentheses is unclear. Are they the intensity ratios of the B-band to the A-band?

Reply: We have modified this sentence to enhance the clarity as below.

“Although the estimated absolute cross section is roughly two times smaller than theoretical predictions, the ratio of the predicted cross sections of the peaks of the B band and the A band, $\sigma(212 \text{ nm})/\sigma(345 \text{ nm}) = 6.6 \times 10^{-18} \text{ cm}^2 / 2.5 \times 10^{-19} \text{ cm}^2 = 26$, is consistent with the observed relative intensity of the peaks (~ 20) as shown in Figure 3.”

8. Line 160: Consider including reference 19 when citing previous calculations related to the excited state dynamics.

Reply: We have cited that reference (Chenet al. *Chemical Physics Letters* **333**, 365–370 (2001)) here, now as citation 20 in the modified version, for comparing the calculated ground state geometry of ClCO. “...a bond angle of 129.1 degrees, consistent with previous calculations.^{20,21,33}”

9. Line 251: I just want to confirm my understanding: Are the blue curves in all four panels of Figure 6 the same as the curve shown in the upper panel of Figure 3 (stitching the green and red curves)?

Reply: The B band profile in Figure 3 alone covers a spectral range of 200-320 nm, which covers the entire window shown in Figure 6. We have modified the figure caption for clarity.

“...were decomposed into the contributions of ClCO formation (blue, see the B band profile in upper panel of Figure 3) and Cl₂ depletion³¹...”

Reviewer #2 (Remarks to the Author):

This is an important paper on the observation of the ClCO radical in the gas phase by its UV-Vis absorption spectrum in conditions relevant to the chlorine chemistry in the Venus mesosphere. The manuscript is very well written and the results presented nicely and in a competent manner. High level ab initio electronic structure calculations are presented and help to understand the photochemistry of this important radical. The discussions and repercussions of the study are enlightening. In my view, the paper can be published as is.

Reply: We much appreciate the support and thank the reviewer for the kind comments.

Reviewer #3 (Remarks to the Author):

Review of The UV–Vis Spectrum of the ClCO Radical in the Catalytic Cycle of the Cl-Initiated CO Oxidation by Chao W., et al.

Summary

This work reports on the UV-Vis spectrum of chloroformyl radicals (ClCO), which are supposed to be an important atmospheric species in the carbon cycle in the upper Venus atmosphere, in the 236-294 K temperature and 50-491 Torr pressure ranges. The work presents ab initio calculations as well as laboratory measurements.

Recommendation

The paper is well-written and provides all the necessary information. I recommend publication with minor changes if the comments listed below are addressed.

Main comments

Line 84: What does T represent? Also, even though the provided value of L indicates it is the path length, I would specify the meaning of those symbols in the text.

Reply: We thank the reviewer for this great suggestion. We have modified it as below to make this sentence clearer.

“However, a large fraction of the 193 nm photons were absorbed by (ClCO)₂, resulting in an estimated transmittance of 18 %, based on the absolute cross section of (ClCO)₂ ($\sigma_{\text{ClCO}_2}(193 \text{ nm})^{27} = 3.8 \times 10^{-18} \text{ cm}^2$) and the path length of the flow reactor, excluding the purge region ($L = 45 \text{ cm}$). This absorption resulted in inhomogeneous radical formation down the length of the flow reactor.”

Line 87: I see the negative values in Figure 2A below 380 nm, not 360 nm.

Figure 2A: What is the grey curve?

Reply: We thank the reviewer for pointing out this vague description. We referred to the negative absorption band near 360 nm, but it really starts below 380 nm. We have changed 360 nm to 380 nm.

“The negative signal below 380 nm was attributed to...”

The gray line in Figure 2A is the estimated depletion signal of (ClCO)₂. We have modified the figure caption (see also response to question from reviewer 1).

“...using 193 nm photolysis of (ClCO)₂. The gray line shows the contribution due to the depletion of (ClCO)₂.”

Line 116: What are the uncertainties on the [Cl]₀ signal intensities?

Reply: We estimated the uncertainty on [Cl]₀, based on the baseline stability, 10%, and the uncertainty in effective absorption pathlength, 10%, i.e., $(10\%^2 + 10\%^2)^{0.5} = 14\%$ as described in the SI, “Error Analysis”.

We have added these error estimates to the main text for convenience.

“..., we normalized the signal intensities to $[Cl]_0 = (11.7 \pm 1.7) \times 10^{13} \text{ cm}^{-3}$, $(10.0 \pm 1.4) \times 10^{13} \text{ cm}^{-3}$ and $(8.9 \pm 1.3) \times 10^{13} \text{ cm}^{-3}$ at 498 Torr, 241 Torr and 91 Torr, respectively.”

However, it is questionable to directly link this uncertainty to the uncertainty of the ClCO absolute cross section because complete scavenging of the Cl atoms is hard to quantify without kinetic analysis.

We have now included text to emphasize this point:.

“...in the 270 – 300 nm window to the B-band profile. We determined the absolute cross section of $\sigma_{ClCO}(223 \text{ nm})$ to be approximately $3 \times 10^{-18} \text{ cm}^2$. This value should be considered as a lower bound since complete scavenging of the Cl atoms is hard to quantify without kinetic analysis. Although...”

Line 126: Could you specify how much the temperature range matches with the literature on Venus's mesospheric temperatures? If I remember correctly, those vary between 150 K and 300 K. Could you add a discussion in the text on how those can be extended to cover the whole useful temperature range? This comment applies to all temperature dependence discussed in the paper...

Reply: The 150-300 K temperature range covers the altitude ranging from 60-90 km, including the upper cloud layer and beyond. The temperature dependence discussed in line 126 supported that both the A band and B band have the same origin, which is the ClCO radical.

We have added text to the introduction to help contextualizing the present work in relation to the Venus

atmosphere. First, we specify the conditions of Venus's mesosphere:

“...chlorine chemistry has been proposed to play a pivotal role in regulating the concentration of molecular oxygen in Venus's mesosphere.⁸ The upper atmosphere of Venus (60-90 km, 300-150K)⁹ is composed primarily of CO₂...”

Then, we discuss its importance in “The Chemical Insights and Potential Implications”.

“...this first direct observation of the ClCO radical in the UV-Vis region provides a new way to determine important parameters related to ClCO with reduced uncertainty at 236 K (e.g. K_{ClCO} and the rate coefficient for the ClCO + O₂ reaction). The ClCO absorption signals near 223 nm over 236 – 294 K were strong, showing that the K_{ClCO} could be determined from the observed CO dependence using Cl₂ as the chlorine precursor over 150 – 300 K for accurately modeling the catalytic effect of the Cl-initiated CO oxidation in Venus's mesosphere. A better estimation of the O₂ profile...”

Finally, we have slightly modified the final sentence in the abstract, to emphasize the aspect of O₂ regulation in Venus' atmosphere. It now reads:

“...the reported spectrum will enable future measurements of essential kinetic parameters related to the ClCO radicals, which are important in regulating the O₂ abundance in Venus's mesosphere.”